# Advances in the In Vivo Molecular Imaging of Invasive Aspergillosis

**DOI:** 10.3390/jof6040338

**Published:** 2020-12-04

**Authors:** Matthias Gunzer, Christopher R. Thornton, Nicolas Beziere

**Affiliations:** 1Institute for Experimental Immunology and Imaging, University Hospital, University Duisburg-Essen, 45147 Essen, Germany; 2Leibniz-Institut für Analytische Wissenschaften-ISAS-e.V., 44227 Dortmund, Germany; 3ISCA Diagnostics Ltd. and Biosciences, College of Life & Environmental Sciences, University of Exeter, Exeter EX4 4PY, UK; c.r.thornton@exeter.ac.uk; 4Werner Siemens Imaging Center, Department of Preclinical Imaging and Radiopharmacy, Eberhard Karls University Tübingen, 72076 Tübingen, Germany

**Keywords:** *Aspergillus fumigatus*, invasive pulmonary aspergillosis, aspergillosis, imaging, positron emission tomography, immunoPET, siderophores, radiolabel

## Abstract

Invasive pulmonary aspergillosis (IPA) is a life-threatening infection of immunocompromised patients with *Aspergillus fumigatus*, a ubiquitous environmental mould. While there are numerous functioning antifungal therapies, their high cost, substantial side effects and fear of overt resistance development preclude permanent prophylactic medication of risk-patients. Hence, a fast and definitive diagnosis of IPA is desirable, to quickly identify those patients that really require aggressive antimycotic treatment and to follow the course of the therapeutic intervention. However, despite decades of research into this issue, such a diagnostic procedure is still not available. Here, we discuss the array of currently available methods for IPA detection and their limits. We then show that molecular imaging using positron emission tomography (PET) combined with morphological computed tomography or magnetic imaging is highly promising to become a future non-invasive approach for IPA diagnosis and therapy monitoring, albeit still requiring thorough validation and relying on further acceptance and dissemination of the approach. Thereby, our approach using the *A. fumigatus*-specific humanized monoclonal antibody hJF5 labelled with ^64^Cu as PET-tracer has proven highly effective in pre-clinical models and hence bears high potential for human application.

## 1. *Aspergillus fumigatus* and Invasive Pulmonary Aspergillosis

*Aspergillus fumigatus* (*A. fumigatus*) is an ubiquitous saprotrophic mould, whose air-borne spores are routinely inhaled [1], and with DNA of the pathogen detectable in bronchoalveolar lavage (BAL) specimens from healthy individuals [2]. Exposure of humans to *A. fumigatus* can result in a number of different syndromes depending on the extent of the host’s immune response. These syndromes range from hypersensitivity/allergy-associated diseases such as allergic bronchopulmonary aspergillosis (ABPA) in immunocompetent individuals to severe invasive disease (invasive pulmonary aspergillosis, IPA) under conditions of severe immunodeficiency [1]. Until the turn of the century, IPA was mainly seen in neutropenic patients, where it was (and remains) the principal life-threatening mould infection of immunocompromised individuals. However, the populations of patients at-risk for the disease have expanded dramatically with the advent of modern tumour therapies such as small molecule kinase inhibitors or chimeric antigen receptor (CAR) T cells, as well as aggressive treatment for graft-versus-host-disease (GVHD) using high-dose corticosteroids [1]. In the case of CAR T cell therapy, the often-required aggressive management of side effects such as cytokine release syndrome or also the lymphodepletion therapy applied before infusion of CARs do render patients highly susceptible to infections including invasive fungal diseases [3]. In addition, severe viral infections of the lung such as influenza [4] or COVID-19 have led to the emergence of life-threatening co-infections with *A. fumigatus* in otherwise immunocompetent individuals [5,6].

## 2. Clinical Diagnosis of IPA

While effective antifungal drugs are available in the clinic, growing azole resistance in clinical strains of *A. fumigatus* is a serious issue for the successful treatment of IPA [7,8,9]. However, with a number of new mould-active compounds on the horizon^8^, treatment per se is arguably not the most pressing issue in disease management. Rather, it is our failure to accurately and rapidly diagnose the disease that is driving sub-optimal therapy^1^. Most recent consensus definitions list a number of diagnostic procedures that are required for definitive identification of IPA. Among them are PCR (also combined with DNA sequencing for species determination), high resolution computed tomography (CT), and galactomannan testing of the peripheral blood or BAL recovered during invasive bronchoscopy. Due to its ease-of-use, throughput, low cost and relative patient comfort, CT has proven to be the preferred method for patient screening and the only imaging method recommended for IPA diagnosis. In addition, histological identification of fungal elements in invasive lung biopsies or fungal culture from blood are considered relevant [10]. However, abnormalities in a chest-CT that might indicate IPA are highly variable and by no means specific, making a definitive diagnosis extremely difficult. Furthermore, the growing numbers of non-neutropenic patients suffering from IPA require adjustments to the existing diagnostic paradigms, since established criteria for neutropenic patients are not necessarily translatable to other patient groups [1]. Galactomannan, the predicate microbiological indicator of IPA, is also problematic in patients receiving prophylactic antifungal treatment and which, along with culture, suffers from slow turnaround times in many centres. Consequently, despite substantial progress in IPA detection, there is still no single gold-standard test for the disease that provides high degrees of sensitivity and specificity, and which is non-invasive. As a result, febrile patients with a fever of unknown origin (FUO) are typically treated empirically first with antibiotics and, in the case of persistent fever and neutropenia, with antifungal therapy some days later [11,12].

## 3. Molecular Imaging as an Add-On to Classical Radiology

Molecular imaging aims to circumvent the limitations of classical, purely anatomical, radiological techniques (X-ray, CT, MRI) by providing information based on signals emitted by reporting agents bound to a tracer (e.g., glucose absorbed by metabolically active tissue) or targeting molecule (e.g., a small molecule or antibody binding specifically to a receptor), with specific biological and pharmacokinetic profiles and binding characteristics [13]. Through high contrast delivered by the reporting agent, these techniques can provide information directly correlated to the patho-physiological state of the organism at the molecular level, information that is not obtainable by anatomical data alone [14]. Currently, two main forms of molecular imaging are used: first, optical imaging, relying on the detection of photons emitted by a luminescent or fluorescent reporter [15]; second, radiation-based imaging, relying on the emission of either positrons (β^+^) [16,17] or γ-rays [18,19] from radioactive isotopes. All of these different reporter types have been investigated in relation to IPA in order to develop tools for both researchers and clinicians to diagnose and study *Aspergillus* infection and to monitor therapeutic interventions in vivo. These approaches to imaging IPA will be reviewed in this article.

## 4. Optical Imaging Enabled by Genetically Modified *Aspergillus* Species

Optical imaging of infections in general, and IPA in particular, stem from three different approaches. First, bioluminescent imaging, based on the autonomous generation of light from luciferin oxidation by the luciferase enzyme [20]. This has been enabled for *Aspergillus* species by inclusion of the luciferase gene into the genome of *A. fumigatus* under the promoter *gpdA* [21]. This tool has been mostly used for monitoring the growth and therapeutic response of *A. fumigatus* infection in vitro and in vivo in animal models, with good sensitivity and high specificity [21,22,23,24] as shown in Figure 1, and could be further enhanced by the use of MRI in multimodal environments [25]. However, the bioluminescent approach is limited not only by the need for luciferin injection but also by the requirement for genetically modified strains, thus restricting bioluminescent imaging to the pre-clinical setting with no possibility for clinical translation.

The first limitation, namely injection of an exogenous compound, has been circumvented by the tailoring of fluorescent strains of *Aspergillus* expressing various fluorescent proteins such as the green fluorescent protein [26]. While the inherent limitations of macroscopic optical imaging (especially penetration depth and low resolution in vivo) have a strong influence on the outcome of the experiments, some interesting results have been produced in vitro, notably on the interaction between *A. fumigatus* and phagocytic and non-phagocytic cells [26]. Other fluorescent variants (fluorescent at other wavelengths) have also been obtained for similar purposes [27]. These investigations were expanded upon by the introduction of red-shifted FLuorescent *Aspergillus* REporter (FLARE) [28], red fluorophore (DsRed)-expressing *Aspergillus* conidia additionally grafted with a fluorophore on its surface, allowing for fungal viability and integrity studies in vivo and thus the interaction of the fungus with the host [29,30]. This new variant enabled several additional pre-clinical studies in vitro and in vivo (reviewed elsewhere [31]) and is still actively used, including its evolution—Mitochondria and FLuorescence *Aspergillus* REporter (MitoFLARE)—that allows real-time investigation of granulocyte killing efficiency on *A. fumigatus* in vitro [32]. Further refinement and use of fluorescent strains, combined with technological advances, have allowed a closer look into IPA and notably of host-*Aspergillus* interactions, for example using endoscopy to visualize host responses in situ and in vivo [33]. Collectively, while remaining highly valuable tools for a better understanding of IPA in pre-clinical settings, genetically modified *Aspergillus* strains cannot provide any direct clinical benefits.

## 5. Scintigraphy and SPECT Imaging of Fungal Infections

Scintigraphy and Single photon emission computed tomography (SPECT) rely on the emission of γ-rays from radiotracers such as ^99m^Tc, ^111^In or ^123^I. The detection of γ-rays by the scintillators, rotated around the sample in the case of SPECT, yield an image of the distribution of the radiotracer in space [18]. In itself, a SPECT image does not contain anatomical information. Hence, a simultaneous CT scan to obtain anatomical information is often performed, providing a reference to locate the radiotracer in the organism. Functional radiotracers require the attachment of γ-ray emitting isotopes onto a chemically modified targeting vector or tracer. In order to provide specificity to the infection site in fungal disease imaging, two main routes have been investigated: radiolabelling of antifungal drugs and radiolabelling of antifungal peptides. More specifically, radiolabelling of the azole antifungal Fluconazole by ^99m^Tc (^99m^Tc-Fluconazole) showed increased accumulation at sites of candidiasis (disseminated disease caused by the yeast pathogen *Candida albicans*), but not in bacterial infection or during sterile inflammation [34]. Unfortunately, ^99m^Tc-Fluconazole showed no specific accumulation during *A. fumigatus* infection, which was not totally unexpected given the poor efficiency of the drug against IPA. A similar approach was used with the polyene antifungal drug Amphotericin B, also using ^99m^Tc as a radiolabel [35], showing promising results with specific accumulation in mould pathogens in in vitro culture models. While these results still require confirmation in vivo, it is foreseeable that the specificity of this radiotracer will be limited to all species sensitive to Amphotericin B and will thus not be *Aspergillus*-specific. A parallel approach focused on improving the delivery of drugs to IPA nodules has highlighted the potential of novel formulations (namely poly(lactide-co-glycolic acid) nanoparticles containing Voriconazole) used in inhalators to deliver drugs efficiently to infected lungs [36]. Radiolabelling of such particles by ^99m^Tc could show their accumulation in various major organs as well as increased accumulation and retention in diseased lungs, which could represent a new theranostic approach. However, as with other drug-based approaches, the specificity toward *Aspergillus* species cannot be guaranteed, and accumulation in other types of infections is highly likely.

In addition to small molecular antifungal drugs being radiolabelled, the same approach was applied to anti-microbial peptides [37] such as ubiquicidin [38] and lactoferrin [39] which exhibit antifungal activity against several genera including *Candida* and *Aspergillus*. Here as well, ^99m^Tc has been the radioisotope of choice due to the chemical ease with which a suitable chelator can be attached to the antibiotic peptides [40]. The resulting radiolabelled peptides, ^99m^Tc-lactoferrin and ^99m^Tc-ubiquicidin, have shown interesting profiles for imaging of candidiasis, IPA and bacterial disease [41], although with no particular specificity. The potential of ^99m^Tc-ubiquicidin for fungal infection in particular has been investigated in clinical trials over a 10-year period, and has shown promising results in infection diagnosis and site mapping, although with no specificity towards IPA, as reviewed elsewhere [42].

In between the synthetic drug and the natural peptide pathway, specific targeting vectors have been developed and radiolabelled as candidates for specific imaging of various infections. Short cyclic peptides in particular show interesting pharmacokinetic and targeting potentials and have thus been of interest for detecting IPA. ^111^In was used to radiolabel c(CGGRLGPFC) [41,43,44], which has shown specific accumulation in the lungs of mice suffering from IPA as seen in Figure 2. Additional work on the same compound revealed the absence of accumulation of the radiotracer in *Geosmithia infections* [45]. Further work is needed to completely validate the specificity of the radiotracer towards *A. fumigatus*, which would then justify first-in-man studies.

## 6. Positron Emission Tomography (PET) Approaches

Unlike SPECT, positron emission tomography (PET) relies on the β^+^ decay of radioactive isotopes. After an initial energy loss, the emitted positron will then encounter an electron, which will result in an annihilation event emitting two photons in approximately opposite directions with respective energies of 511 keV [46]. Detection of these photons by scintillators under certain conditions (most notably simultaneously) will be registered by the system, allowing for reconstruction of an image representing the distribution of photon-producing events in the sample. Given the negligible ambient β^+^ background, a PET image can thus be directly correlated with the distribution of the radioisotope in the sample, classically-administered as a radiolabelled metallic complex decorating a large molecule (e.g., a peptide or protein) [17] or a radiolabelled organic small molecule [47]. As the decay time is known for the injected isotopes (with the common isotopes and their half-lives being ^11^C, t_1/2_ = 20 min; ^68^Ga, t_1/2_ = 67.7 min; ^18^F, t_1/2_ = 110 min; ^64^Cu, t_1/2_ = 12.7 h and ^89^Zr, t_1/2_ = 78.4 h), decay-corrected signals give a direct insight into the local concentration and the biological properties of the injected radiotracer.

Since the advent of the first PET system, research of targeted radiotracers has been active in a wide variety of fields, most notably cancer. The most broadly used tracer for PET imaging is ^18^F-fluorodeoxyglucose (^18^F-FDG), which is taken up by cells due to the Warburg effect and is the current gold standard to map tumours or their metastases in cancer patients [48] and can be used for brain activity monitoring [49], inflammation and infection [50,51,52] and in general to monitor (patho)-physiological events with a large cellular glucose uptake.

### 6.1. Glucose Uptake

As a result, ^18^F-FDG has been used to monitor infections based on the local increased glucose consumption from both the immune system and proliferating infectious pathogens. Despite the clinical potential of ^18^F-FDG for monitoring therapy after diagnosis of fungal infections, as reviewed elsewhere [53], in the case of IPA, ^18^F-FDG faces three major limitations as a diagnostic procedure: first, it is not specific for the fungus but rather highlights the immune response to an infection and general cellular activity; second, ^18^F-FDG is readily taken up by active muscle cells, providing a variable but significant background signal, notably from the heart [54]; third, ^18^F-FDG will provide “false positive” signals as it is enriched by other lung diseases such as lung tumours [55], lung fibrosis [56] and various other ailments [57]. This has been shown in pre-clinical models in several studies, notably by comparing the ^18^F-FDG signal in IPA models with other sterile or bacterial infection models, as well as in other fungal infections [58]. In the clinic, ^18^F-FDG has been the topic of numerous investigations regarding its potential to support IPA diagnosis [59,60,61,62]. ^18^F-FDG-driven diagnosis has proven an interesting tool for disease management through therapy monitoring or tracking of secondary invasion sites and is starting to find its place alongside CT as a radiological workhorse for IPA diagnosis as can be seen in Figure 3. Its use is undoubtedly aided by its FDA approval, low radiation exposure of patients due to short half-life, and general availability in all centres equipped with PET scanners.

### 6.2. Radiolabelled Drugs and Metabolic Markers

Despite this, it is evident that ^18^F-FDG PET imaging does not provide any information on the type of infection, whether bacterial, viral or fungal, which has highlighted the limits of this approach in terms of diagnosis and therapy monitoring in the fields of inflammation and infection biology. For this reason, other radiotracers have been developed which aim to distinguish between infectious organisms and host cellular activity. The standard approach remains the radiolabelling of readily available drugs, but the difficulty of the chemistry involved and the lack of disease-specificity render this approach unattractive. Despite these limitations, some interesting results have been published, notably radiolabelling of Fluconazole with ^18^F, a similar approach to ^99m^Tc radiolabelling for candidiasis diagnosis [63,64]. Here again, the choice of Fluconazole as a targeting vector limits its applicability to IPA diagnostics. Other approaches have harnessed the requirements of infectious organisms for specific nutrients in an attempt to increase the specificity of the radiotracer towards the cause of the infection, while diminishing the background signal from healthy tissue. Here, ^18^F-fluoromaltotriose, targeting the maltodextrin transporter of gram-positive and gram-negative bacterial cells [65] or 2-^18^F-fluorodeoxysorbitol [66,67] to image gram-negative bacteria, have been tested. Fungi metabolise maltose and sorbitol differently than bacteria, so the applicability of this approach to fungal imaging is unclear.

### 6.3. Siderophores as Radiometal Chelators

All organisms are strictly dependent on iron, namely Fe^2+^ and Fe^3+^, for growth and survival [68,69,70]. For this reason, tight regulatory mechanisms for iron transport and the control of its oxidative status are active at the cellular and organism level. During an infection, invading microorganisms must take up iron in order to multiply. However, the metal is usually tightly sequestered by the host organism as an anti-infectious strategy [71]. As a result, pathogens have evolved iron “traps” as a way to circumvent the lack of iron during disease propagation. *A. fumigatus* in particular developed two iron scavenging pathways: reductive iron assimilation and siderophore-mediated iron acquisition [72,73]. Both are induced by iron starvation, and the importance of these systems has been shown by genetic de-activation of iron scavenging in in vivo models, although only siderophore biosynthesis appears to be essential to *A. fumigatus* pathogenicity [74]. Siderophores are potent ferric iron chelators and are secreted by most bacterial and fungal pathogens. *A. fumigatus* secretes two cyclic siderophores, fusarinine C (FsC) and its tri-acetylated derivative triacetylfusarinine C (TAFC) [75], which are taken up via siderophore-iron transporters (SITs) after successful chelation of ferric iron. Interestingly, SITs are only present in fungi, and while in certain species SITs can demonstrate substrate-specificity, others can import a broad range of siderophores [76]. For these reasons, building contrast agents or radiotracers around siderophore scaffolds to specifically image infection has been investigated heavily in molecular imaging [77] in both optical imaging [78], using fluorophore conjugation, and PET imaging, using an iron-mimicking radionuclides such as ^68^Ga [78,79,80,81,82] or ^89^Zr [83]. These radiometals can be readily obtained from a Ge/Ga generator for ^68^Ga or are commercially available for ^89^Zr. With a t_1/2_ of 68min, ^68^Ga represents an attractive option for clinical applications as it yields a relatively low radiation burden for the patient, while ^89^Zr allows for longer imaging times (t_1/2_ = 3.27 days) at the cost of a higher radiation burden. This approach of radiolabelling siderophores using ^68^Ga or ^89^Zr have shown promising results in in vivo rat models of IPA in a PET/CT (Figure 4), with high accumulation of the radiotracer in the infected lungs and a renal excretion route alongside high metabolic stability, and no marked accumulation in sterile inflammation or cancer [79,81,84]. However, species-specificity is limited by the choice of siderophore: TAFC-based complexes can be taken up by various other fungal pathogens (such as *Fusarium solani*), but not by other fungi in the same or different genera (*A. terreus*, *A. flavus*, *C. albicans*) or bacteria (*S. aureus*, *Pseudomonas aeruginosa*), despite their clinical importance. As a result, it appears that naturally occurring siderophores lack the specificity required for accurate clinical diagnosis of infectious diseases such as IPA, and chemically engineered siderophores with higher specificity are required to unlock the clinical potential of this approach. Interestingly, chemical approaches including a fluorophore in addition to the radiolabel have been recently published [78] and could represent an approach for initial radiological diagnosis based on PET followed by clinical resection of the fungal mass guided by optical imaging during surgery.

### 6.4. Radiolabelled Antibodies

The use of antibodies able to specifically recognize *Aspergillus* species and to differentiate between actively growing hyphae and inactive spores has recently proved to be a valuable approach for pre-clinical diagnosis of IPA using molecular imaging. The use of radiolabelled antibodies for PET imaging (so called immunoPET [85]) is a relatively new field that has historically focused on oncology, despite its potential for specific diagnosis of different infectious aetiologies that express unique antigens [86]. For IPA specifically, the murine monoclonal antibody (mAb) mJF5 [87] and its humanized counterpart hJF5 [88] have been developed by us. JF5 binds to galactofuranose-rich glycoproteins expressed by the hyphae of all clinically relevant *Aspergillus* species (*A. fumigatus*, *A. terreus*, *A. niger*, *A. flavus* and *A. nidulans*) [89,90] and forms the basis of a novel lateral -flow device (LFD) for IPA detection [87,91,92]. During further investigations by targeted deletion of the gene encoding the enzyme UDP-galactopyranose mutase, JF5 has been shown to bind to the epitope β1,5-galactofuranose [88]. This epitope is absent in mammals, thus preventing non-specific binding of JF5 to host tissues [93,94]. Radiolabelling of mAb mJF5 with ^64^Cu and its use in vivo has demonstrated its ability to differentiate IPA from sterile inflammatory responses or bacterial infections, unlike other approaches published so far. Initially, using mAb mJF5 and the ^64^Cu chelator DOTA, we showed markedly increased uptake of the radiotracer in the lungs of *A. fumigatus*-infected animals when compared to healthy animals, but with accompanying uptake of the tracer by the liver [95] (Figure 5). While the high accumulation of the mJF5-based radiotracer in the lungs is directly linked to the presence of invasive hyphae, the liver signal likely resulted from poor stability of the DOTA chelator and/or hepatic clearance of the radiolabelled antibody. Further optimization of the radiotracer in a subsequent study highlighted the benefit of using NODAGA over DOTA as a chelator, leading to improve in vivo stability and enhanced lung targeting [88].

The antibody JF5 has been validated for in vitro diagnosis of IPA using bodily fluids (serum or BAL fluid) [87,91,92]. A potential translation of the in vivo PET imaging approach to the clinical setting required humanisation of the antibody. This step was performed using complementarity-determining regions (CDR) grafting of the mouse variable heavy (V_H_) and variable light (V_L_) chain CDRs onto a human IgG1 framework, resulting in the hJF5 antibody that displays improved affinity for the Gal*f* epitope compared to its mouse counterpart [88]. Radiolabelling through NODAGA conjugation and ^64^Cu chelation enabled comparative experiments in our neutropenic murine model of IPA. As such, this approach holds enormous promise for specific imaging of IPA in a clinical setting and may even be applicable to other *Aspergillus* diseases such as cerebral aspergillosis, although penetration of the full length antibody into the brain will likely be reliant on disruption of the blood–brain barrier during disease progression [96]. In addition to PET imaging, the JF5 antibody has also very recently been used to obtain high resolution 3D light-sheet fluorescence microscopy images of infected mouse lungs ex vivo by using a fluorophore label, allowing novel insights into different modes of immunosuppression on IPA progression [97].

While currently the only non-invasive IPA-specific imaging procedure, several limitations may hinder the clinical application of radiolabelled hJF5 for PET imaging. Firstly, the pre-clinical models used to test imaging approaches often present a significantly faster disease progression than in humans [98,99]. This limits the window of investigation for the early diagnostic capacity of the imaging method chosen. Secondly, the animal models often rely on complete invasion of the lung by the fungus, whereas the human disease is often represented by nodules of moderate size (a few centimetres) on first appearance of symptoms [100,101]. Lastly, IPA often overlaps with a myriad of other lung diseases such as chronic obstructive pulmonary disease or cystic fibrosis, co-morbidities that are often difficult to investigate pre-clinically. Notwithstanding these uncertainties, immunoPET imaging based on mAb JF5 represents the most promising candidate for clinical translation to date due to its excellent specificity, overall performance in pre-clinical trials and first-in-human compassionate use which commenced in 2018 with promising preliminary results (unpublished).

## 7. Perspectives

Molecular imaging is showing enormous promise for the clinical diagnosis of IPA [1]. While the more general inflammatory conditions surrounding infections imaged by ^18^F-FDG or accumulation of radiolabelled antifungals do not yield the required specificity, siderophore imaging and especially immunoPET have been shown to provide a definitive diagnosis without the need for invasive procedures. Future developments in immunoPET need to address the speed of enrichment in the target tissue and to provide a higher signal-to-noise ratio in acquired images, e.g., by developing smaller targeting vectors such as nanobodies [102]. In any case, the need for rapid and unequivocal diagnosis has lost nothing of its urgency, especially in the current COVID-19 crisis where growing numbers of critically-ill patients are being treated with anti-inflammatory corticosteroids [103], which is opening the door to opportunistic co-infections by *A. fumigatus*.

## Figures and Tables

**Figure 1 jof-06-00338-f001:**
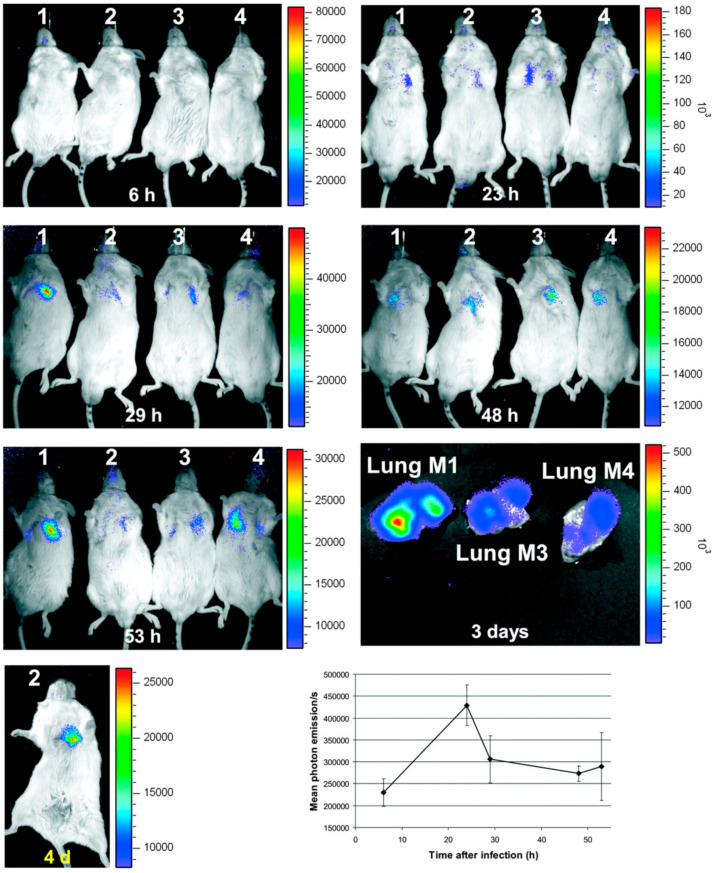
Bioluminescence imaging of aspergillosis. Evolution of luminescence emission from BALB/cJ mice after intra-nasal infection with 2 × 10^6^ conidia of bioluminescent *A. fumigatus*. Emission of light peaked 23 h after injection before declining continuously until sacrifice. Ex vivo images were obtained directly after the last in vivo time point. The lower right graph shows the mean signal from the chest of mice through the experiment. Adapted with permission from Brock et al., *Appl. Environ. Microb.* 2008, *74*, 7023–7035.

**Figure 2 jof-06-00338-f002:**
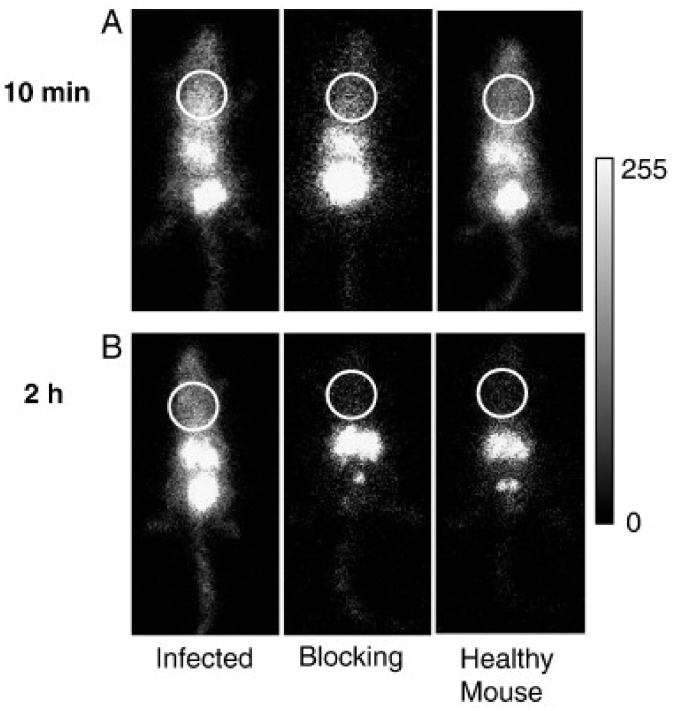
Preclinical γ-scintigraphic imaging of aspergillosis. Images of an infected mouse, a mouse co-injected with unlabelled peptide and a healthy control mouse, acquired (**A**) 10 min and (**B**) 2 h after intravenous injection of 111In-DTPA-c(CGGRLGPFC)-NH2. Circles indicate the chest area encompassing the lungs. Adapted with permission from Yang et al. *Nucl. Med. Biol.* 2009, *36*, 259–266.

**Figure 3 jof-06-00338-f003:**
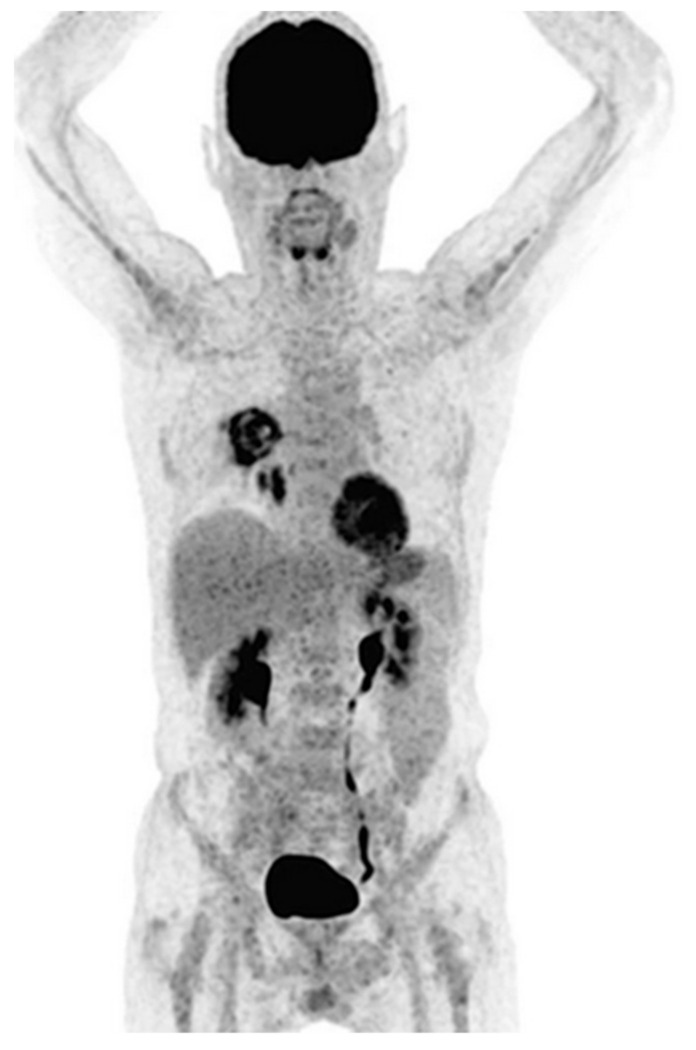
Clinical ^18^F-FDG PET imaging of an acute myelogenous leukaemia patient diagnosed with pulmonary aspergillosis. Complex heterogeneous lesions can be seen in the lungs. Patient was diagnosed using BAL. Adapted from Ankrah et al. *Eur. J. Nucl. Med. Mol. I* 2019, *46*, 174–183.

**Figure 4 jof-06-00338-f004:**
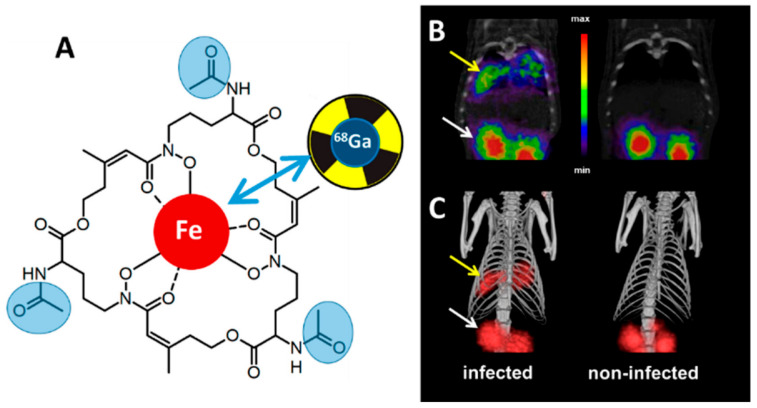
Radiolabelled siderophore imaging of Aspergillosis in a rat model. (**A**) Possible chelation of Fe(III) or ^68^Ga by Triacetylfusarinine C (TAFC). Possible chemical modification sites are highlighted in blue. (**B**) µPET/Computed Tomography coronal slices and (**C**) volume rendered 3D images rat infected by *A. fumigatus* after injection of [^68^Ga]Ga-TAFC showing accumulation in the lungs (yellow arrow) and excretion through the kidneys (white arrow) of the radiotracer. Adapted from Petrik et al. *J. Fungi* 2020, *6*.

**Figure 5 jof-06-00338-f005:**
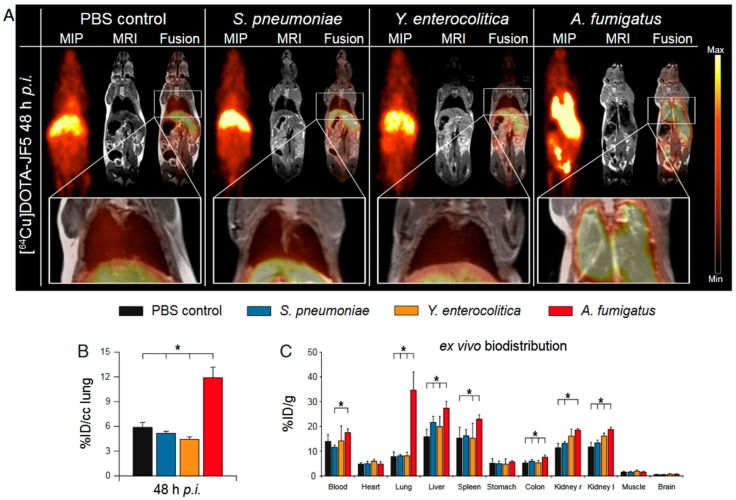
Preclinical JF5-based PET imaging of aspergillosis in mice. (**A**) [^64^Cu]Cu-DOTA-mJF5 PET coronal maximum intensity projection (MIP) and magnetic resonance images (MRI) overlayed (Fusion) in control (PBS inoculated), *S. pneumoniae*, *Y. enterocolitica* and *A. fumigatus* lung infections. Close-up of the lung area in the insert below. (**B**) Quantification of in vivo radiotracer accumulation in the lungs 48 h post injection (*p.i.*). (**C**) Quantification of radiotracer bio-distribution ex vivo in excised organs. Data are expressed as the average ± standard deviation of the injected dose (ID), one-way ANOVA, * *p* < 0.05. Adapted from Rolle et al. *Proc. Natl. Acad. Sci. USA* 2016, *113*, E1026–E1033.

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
