# Peer review of "Advances in the In Vivo Molecular Imaging of Invasive Aspergillosis"

_jof, 2020, doi:10.3390/jof6040338_

Round 1

Reviewer 1 Report

This Review paper deals with methods of radiologic molecular imaging technology to advance the diagnostic capabilities for invasive aspergillosis. I agree with the authors that improvement in clinical diagnostics for invasive aspergillosis are urgently needed. The paper addresses the current State of the Art and provides a synopsis of the  experimental radiologic techniques that may be clinically useful.

The paper is well written, well organized, , and easy to follow particularly for the non-radiology audience.

I have a few minor suggestions/issues-

1-Lines 46,47- Azole resistance in Aspergillus- refs provided #6,7 are not appropriate. Pl provide better references.

2-lines 64,65- the empiric approach of antifungal therapy is in the neutropenic setting. Pl clarify.

3-lines- 186-189- 'PET has started to replace CT as the radiological workhorse'. I don't believe so. Most centers are not using pet scans for diagnosis- PET has not become a part of the Guidelines, procedure is expensive, not widely accepted/validated, not readily available in many centers. Pl change accordingly and also emphasise throughout the text and the abstract.

Author Response

The paper is well written, well organized, and easy to follow particularly for the non-radiology audience.

We thank the reviewer for the kind feedback. Our point-by-point response can be seen below, and changes in the manuscript addressing the specific points have been made using Track Changes.

I have a few minor suggestions/issues-

1-Lines 46,47- Azole resistance in Aspergillus- refs provided #6,7 are not appropriate. Pl provide better references.

We have now added two references (Verweij et al. 2016 Clin. Infect. Dis. ; Chowdhary et al., 2017  J. Infect. Dis.) dealing with azole resistance, focusing on the resistance mechanisms, the therapeutic options available and the significance for human medicine and agriculture, replacing Zoran et al. 2018 Front. Microbiol. which was mistakenly added. We believe however that Patterson et al. 2016, Clin. Infect. Dis. provides a good reference frame to the reader regarding clinical routine.

2-lines 64,65- the empiric approach of antifungal therapy is in the neutropenic setting. Pl clarify.

We have clarified the sentence L70-72.

3-lines- 186-189- 'PET has started to replace CT as the radiological workhorse'. I don't believe so. Most centers are not using pet scans for diagnosis- PET has not become a part of the Guidelines, procedure is expensive, not widely accepted/validated, not readily available in many centers. Pl change accordingly and also emphasise throughout the text and the abstract.

We have reworded the sentence so as to not mislead the reader in believing FDG-PET is replacing CT in routine examination. We have in addition emphasized the need for further validation in several instances in the text. (lines 58-60 and 196-197).

While we agree that the procedure, be it with FDG or other radiotracers, still has to be thoroughly validated and PET might not be as disseminated as CT or even MRI, costs per scan are not prohibitive. In the case of critically ill patients that might or might not benefit from the administration of expensive and side-effect heavy drugs, the cost of a conclusive PET scan will be negligible compared to the cost of a wrong therapeutic choice.

Reviewer 2 Report

this paper provides a clear and profound overview of the developments on medical imaging of IPA. 

there are few items that need to be addressed

line 40 in my opinion CAR-T cell treatment as such is not the problem, the patients having specific (blood) cancers like leukemia are at risk; the authors should explain this somewhat more clear why they see CAR-T therapy here a risk factor

line 223: these specific transporters are called "siderophore-iron transports (SITs)

line 225: this part is not completely correct: siderophores are imported via SITs, some fungal species have SITs with broad substrate specificity like the one in Candida while other SITS have clear substrate specificity: see ref 73
the transporters (SITs) are not internalized but the siderophore-Fe complex is

line 254: peptidoglycan molecules is incorrect, those are specific bacterial structures and not present in fungi. The authors mean galactomannan which can be associated to cell wall polysaccharides glucan
authors can refer to
https://academic.oup.com/mmy/article/47/Supplement_1/S104/1069850

minor

remove italics in line 152-153

check paper for introduction of abbreviations like FDG and CDR

The authors do not refer to Figure 5

Author Response

this paper provides a clear and profound overview of the developments on medical imaging of IPA. 

Thank you for the kind feedback. Please find our point-by-point response below in red and the changes made to the manuscript using Track Changes.

there are few items that need to be addressed

line 40 in my opinion CAR-T cell treatment as such is not the problem, the patients having specific (blood) cancers like leukemia are at risk; the authors should explain this somewhat more clear why they see CAR-T therapy here a risk factor

We have detailed this now more in the text (lines 46-49). CAR-T cell treatment often requires intensive care management of cytokine release syndromes and neurotoxicity which can include high-dose steroids or anti IL6R treatment. This treatment can generally increase the infection risk with IFA being particularly increased (personal communication with our hematology ward). Hence, the management required to allow the survival of a CAR T-cell therapy can increase the risk of infection. We have also added a new reference (new #3) to support this notion.

line 223: these specific transporters are called "siderophore-iron transports (SITs)

This is now mentioned in the text.

line 225: this part is not completely correct: siderophores are imported via SITs, some fungal species have SITs with broad substrate specificity like the one in Candida while other SITS have clear substrate specificity: see ref 73 the transporters (SITs) are not internalized but the siderophore-Fe complex is

Thank you for the clarification. We have reformulated the sentences L233-238 regarding SITs and their specificity to hopefully lift any confusion.

line 254: peptidoglycan molecules is incorrect, those are specific bacterial structures and not present in fungi. The authors mean galactomannan which can be associated to cell wall polysaccharides glucan authors can refer to https://academic.oup.com/mmy/article/47/Supplement_1/S104/1069850

Thank you for the correction. We meant glycoprotein and have corrected our mistake, and referred to the citation provided by the reviewer in the text (line 266).

minor

remove italics in line 152-153

Done (now line 161/162)

check paper for introduction of abbreviations like FDG and CDR

The first instance of FDG in the text is now written in full (18F-fluorodeoxyglucose), and its abbreviation homogenized to 18F-FDG. We also have been through the text carefully again to introduce all abbreviations in their order of appearance.

The authors do not refer to Figure 5

This omission has been corrected L276.